# ANATOMY-AWARE REPRESENTATION LEARNING FOR MEDICAL ULTRASOUND

**Seok-Hwan Oh**
Barreleye Inc.
Seoul, Korea

**Myeong-Gee Kim**
Barreleye Inc.
Seoul, Korea

**Guil Jung**
KAIST
Daejeon, Korea

**Hyeon-Jik Lee**
KAIST
Daejeon, Korea

**Young-Min Kim**
KAIST
Daejeon, Korea

**Sang-Yun Kim**
KAIST
Daejeon, Korea

**Hyuksool Kwon**
SNUBH
Seongnam, Korea

**Hyeon-Min Bae**[*]
KAIST
Daejeon, Korea

## ABSTRACT

Diagnostic accuracy of ultrasound imaging is limited by qualitative variability and its reliance on the expertise of medical professionals. Such challenges increase demand for computer-aided diagnostic systems that enhance diagnostic accuracy and efficiency. However, the unique texture and structural attributes of ultrasound images, and the scarcity of large-scale ultrasound datasets hinder the effective application of conventional machine learning methodologies. To address the challenges, we propose Anatomy-aware Representation Learning (ARL), a self-supervised representation learning framework specifically designed for medical ultrasound imaging. ARL incorporates an anatomy-aware Vision Transformer (A-ViT). The A-ViT is parameterized, using the proposed large-scale medical ultrasound dataset, to provide anatomy-aware feature representations. Through extensive experiments across various ultrasound-based diagnostic tasks, including breast and thyroid cancer, cardiac view classification, and gallbladder tumor and COVID-19 identification, we demonstrate that ARL significantly outperforms existing self-supervised learning (SSL) baselines. The results demonstrate the potential of ARL in advancing medical ultrasound diagnostics by providing anatomy-specific feature representation.

## 1 INTRODUCTION

Medical ultrasound (US) is one of the most widely used medical imaging modalities. US is a preferred modality for the early diagnosis of various diseases due to its non-ionizing, cost-effective, and real-time nature. Recently, the application of artificial intelligence (AI) in medical diagnostics has increased significantly (Kumar et al. (2023)). Advancements in object recognition, segmentation, and image classification technologies over the past few years have enabled more accurate and efficient detection and diagnosis of disease, including lung disease diagnosis from computed tomography (CT) scans and breast tumor identification in mammography (Suganyadevi et al. (2022)). Medical US is often characterized by limited image quality and significant qualitative variations, making diagnostic accuracy heavily reliant on the expertise of medical professionals (Oelze & Mamou (2016)). Such reliance has driven a growing demand for computer-assisted diagnosis systems capable of providing quantitative assessments in medical US and enhance diagnostic precision (Akkus et al. (2019); Wang et al. (2021)).

However, the availability of medical US data is limited due to several challenges, including the high cost and logistical complexities of acquiring annotated datasets, as well as privacy concerns and regulatory restrictions in medical imaging (Paleyes et al. (2022); Mazurowski et al. (2019)). Compared to computer vision and natural language processing (Deng et al. (2009); Kirillov et al. (2023); Zhou et al. (2017)), the number of available medical US datasets is considerably smaller. For instance, the open-access breast US image dataset (Al-Dhabyani et al. (2020); Gómez-Flores et al. (2024))

---
[*]Corresponding author

and the thyroid US dataset (Pedraza et al. (2015a)) contain less than a thousand annotated images. The scarcity of data presents significant challenges in training AI models, as their performance and generalizability are constrained. US image attributes are affected by variations in equipment specifications, imaging techniques, and probe characteristics, which makes them particularly prone to the domain shift problem (Ortiz et al. (2012)). As a result, a model trained on the dataset with limited diversity may perform poorly on another dataset collected under different conditions.

In computer vision, self-supervised visual representation learning has demonstrated its effectiveness in capturing general image attributes. Pre-trained models generated from the representation learning are then utilized for downstream tasks, achieving robust performance even with limited data. Recent advancements, such as DINO (Caron et al. (2021)) and MAE (He et al. (2022)), have introduced self-supervised learning methodologies for natural images (NI), demonstrating their effectiveness across a wide range of NI downstream tasks. However, we observed that pre-trained models based on NI datasets fail to perform effectively on US-related downstream tasks. Such performance gap arises from the substantial differences in image attributes between NIs and medical US images, which vary significantly in terms of texture, noise, and structural characteristics. For instance, medical US images typically exhibit a speckled texture caused by the physical properties of sound waves interacting with tissues, which is absent in NIs (Damerjian et al. (2014)).

To address the challenge, we propose a visual representation learning scheme designed for medical US. In this study, we have curated one of the largest-scale datasets dedicated to medical US, which serves as the foundation for the proposed representation learning. Medical US images present distinct attributes that vary significantly depending on the organ being imaged. To address the challenges, we introduce Anatomy-aware Representation Learning (ARL), a framework that utilizes an anatomy-conditioned deformable transformer (ACDT) to ensure that representation learning is tailored to the specific organ or anatomical region being imaged. Through extensive experiments, we demonstrate that ARL effectively facilitates representation learning, enabling application across a wide range of medical US tasks.

## 2 RELATED WORK

**Medical Ultrasound.** Medical US is an essential diagnostic tool for a variety of medical conditions, including breast, thyroid, and gallbladder cancer, and cardiac and lung disease. In breast US, real-time observation of breast masses (Cao et al. (2017)) and identification of malignancy (Cheng et al. (2010)) are increasingly supported by AI-based techniques. The AI-driven methods not only reduce operator dependency but also significantly improve diagnostic accuracy (Shen et al. (2021)). Similarly, thyroid US plays a pivotal role in detecting thyroid cancer (Wang et al. (2019)). The use of learning-based approaches to analyze the radiological features of thyroid tumors and determine the malignancy has grown, leading to enhanced diagnostic precision and reliability. US is widely applied in cardiac care due to its ability to provide real-time imaging of the beating heart. In echocardiography (Echo), deep learning models are employed to classify echocardiographic views and evaluate cardiac functionality, enabling more efficient and accurate cardiac assessments (Madani et al. (2018)). During the COVID-19 pandemic, medical US emerged as a critical tool for detecting lung abnormalities. In particular, the identification of abnormal symptoms in lung US was increasingly facilitated by machine learning–based approaches (La Salvia et al. (2021)). Additionally, US plays a significant role in detecting gallbladder tumors, supporting the early-stage identification of tumors, and improving clinical outcomes (Basu et al. (2024)).

**Self-supervised learning (SSL).** SSL methods are designed to learn meaningful representations by leveraging supervisory cues derived from unlabeled data. Instance discrimination is one of the leading paradigms, which enforces consistency across augmented views of an image. Instance discrimination includes contrastive learning and non-contrastive learning. In contrastive learning, methods such as SimCLR (Chen et al. (2020)) and MoCo (He et al. (2020)) learn to discriminate positive pairs from negative pairs. The objective is to bring similar representations closer together while pushing dissimilar ones apart. However, the approaches necessitate the use of a large number of negative pairs, which limits the widespread use of the frameworks. In contrast, non-contrastive learning methods, including BYOL(Grill et al. (2020)), SimSiam (Chen & He (2021)), and DINO, propose to maximize the similarity between augmented views without relying on negative samples. Notably, DINO introduces a self-distillation mechanism via a matching loss, which has demonstrated

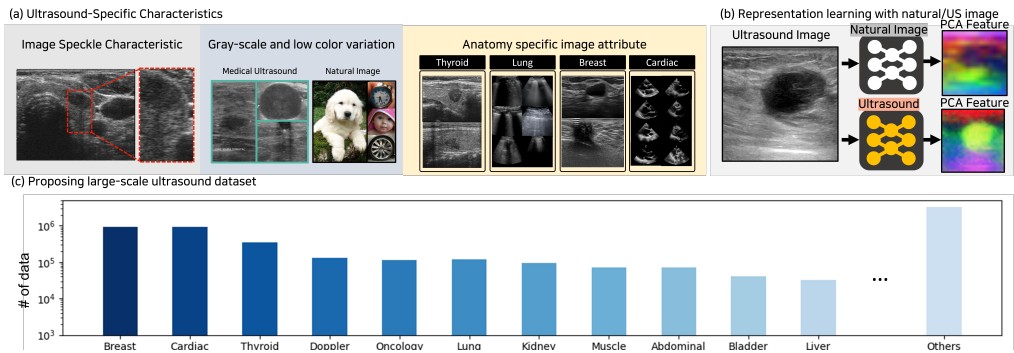

Figure 1: (a) Difference between medical US and natural images. (b) PCA of the features using Dino v3 and the proposed scheme.(c) Distribution of the proposed large-scale US dataset.

significant potential across a variety of computer vision tasks. In addition, masked autoencoders (MAE) have emerged as another significant approach for SSL. Inspired by SSL methodologies in natural language processing (NLP), MAE learns representations by masking portions of an image and training the model to reconstruct the missing content (He et al. (2022); Feichtenhofer et al. (2022)). MAE representation learning encourages the model to extract meaningful features from the contextual information provided by the unmasked regions, further enhancing its ability to learn robust representations.

**Self-supervised learning in medical image.** In the medical domain, there is a growing trend toward building modality-specific foundation models via SSL on large scale medical datasets (Kang et al. (2023b)). Kang et al. (2023a) assembled an extensive pathology image repository and proposed a domain aligned representation learning framework that surpassed traditional baselines on multiple downstream pathology tasks. Likewise, the MIS-FM model (Wang et al. (2023)), pre-trained on three-dimensional computed tomography datasets, achieved state-of-the-art multi organ segmentation performance. Together, these studies highlight the potential of SSL-derived foundational models when adapted to the specific characteristics of each imaging modality. As AI applications in the US domain continue to expand, the development of a US foundation model, capable of supporting a broad spectrum of downstream tasks, will facilitate the delivery of higher quality AI assisted healthcare.

**Self-supervised learning with anatomical features.** Recently, there has been growing interest in SSL frameworks that incorporate anatomical characteristics observed in medical images. Jiao et al. (2020) propose a self-supervised framework that leverages fetal ultrasound videos to capture anatomical characteristics inherent to the fetal imaging domain. Fu et al. (2022) further exploit fetal-specific anatomical cues by constructing anatomy-aware pretext objectives to enhance representation learning within the domain. Hu et al. (2022) introduce a methodology that enforces anatomical consistency across multi-modal medical data, thereby preserving shared structural information across modalities. While these approaches demonstrate the value of incorporating anatomical cues into SSL, they are primarily designed for a single anatomical domain. However, medical ultrasound exhibits anatomical heterogeneity, where diagnostically meaningful features vary substantially across organs. To obtain generalizable representations applicable across anatomy categories, we propose an SSL framework that learns anatomy-specific feature attributes over a wide range of organs, accounting for the inherent variability of medical ultrasound.

## 3 METHOD: ANATOMY-AWARE REPRESENTATION LEARNING

### 3.1 DIFFERENCE FROM NATURAL IMAGES

SSL has primarily been developed and evaluated using large-scale NI datasets. However, medical US images exhibit unique characteristics that significantly differ from those of NIs. Consequently, as demonstrated in Fig.1(b), SSL models trained on NI datasets perform poorly when applied to US

imaging tasks. This section presents the key features that distinguish US images from general NIs, demonstrating the challenges associated with transferring SSL to the medical US domain.

**Image speckle characteristics.** One of the notable differences between medical US and NIs lies in their speckle characteristics. In NIs, adjacent pixels generally exhibit minimal variance due to smooth gradients and consistent textures. In contrast, US images are dominated by significant speckle noise. The noise arises from the interaction of sound waves with tissue structures, producing granular patterns throughout the image (Matrone et al. (2014)). Speckle patterns are not merely artifacts but are often leveraged as essential features in diagnostic applications, such as assessing tumor malignancy (Tsui et al. (2010)) or evaluating cardiac functionality (Mondillo et al. (2011)). However, SSL methods trained on speckle-free NIs tend to diminish the importance of speckle-related features, leading to suboptimal performance on US-specific downstream tasks. Such limitations demonstrate the necessity of developing SSL approaches tailored specifically to medical US data.

**Gray-scale and low color variation.** Another key difference lies in the limited color variation of US images. Unlike NIs, US images are mainly gray-scale with low pixel intensity variation. Such constrained color and intensity range poses additional challenges when applying models trained on NI datasets to US data, as these models are often optimized to leverage the rich color variations that are absent in US images.

**Anatomy-specific image attribute.** Medical US images are captured for diagnostic purposes and are typically examined on the related human organs. Consequently, the images are inherently influenced by the anatomical boundary condition and structural characteristics of the targeted organ. The meaningful feature of medical US images varies substantially depending on the organ being examined, as each organ and lesion type exhibits histologic heterogeneity, reflecting its unique anatomical and functional characteristics. Therefore, it is essential to design representation learning methods that are tailored to the specific organ being imaged to ensure meaningful and accurate feature extraction. To address the challenge, we propose ARL, a novel approach that incorporates the anatomical context of the imaged organ into the learning process. The proposed framework aims to improve both the accuracy and generalizability of self-supervised models, thereby advancing their applicability in medical US imaging tasks.

## 3.2 LARGE-SCALE MEDICAL ULTRASOUND DATASET FORMATION

Effective SSL relies on large-scale datasets to enable neural networks to learn robust and generalizable features. To facilitate the development of the proposed ARL framework, we configured a comprehensive medical US dataset comprising 5.2 million images aggregated from diverse sources, including publicly available datasets and multiple medical institutions. Specifically, the dataset incorporates data from 11 publicly available datasets, as well as US images from 12 medical institutions in the United States, 2 in South Korea, and 1 in India. The detailed description of the dataset is introduced in Appendix A. The dataset is organized into 16 distinct anatomical categories (Fig. 1(c)). The dataset includes images acquired using linear, convex, and sector array probes. The image pixel resolution ranges from 64×64 to 1280×960, with the height and width distributions characterized by $503.1 \pm 167.1$ and $655.8 \pm 238.2$ (mean ± std), respectively. The dataset covers a broad span of imaging depths up to 24 cm. By aggregating a diverse and extensive collection of US data from different geographic regions and clinical conditions, the dataset captures a wide spectrum of imaging variations. Such diversity ensures a more representative and comprehensive dataset, significantly enhancing the potential for effective SSL in medical US applications.

## 3.3 ANATOMY-AWARE VISION TRANSFORMER (A-VIT)

In this work, we propose the A-ViT, a model designed to provide anatomy-adaptive SSL for US images by incorporating anatomical context. A-ViT extends the standard ViT (Dosovitskiy (2020)) by integrating an *anatomy-conditioned deformable transformer*, which adapts feature extraction to the unique characteristics of specific anatomical regions, enabling context-aware representations.

As illustrated in Fig. 2, the model begins by processing the spatial US image through standard patch embedding. This is followed by a series of transformer blocks that perform feature extraction and representation learning. To account for the unique anatomical structures that vary across different

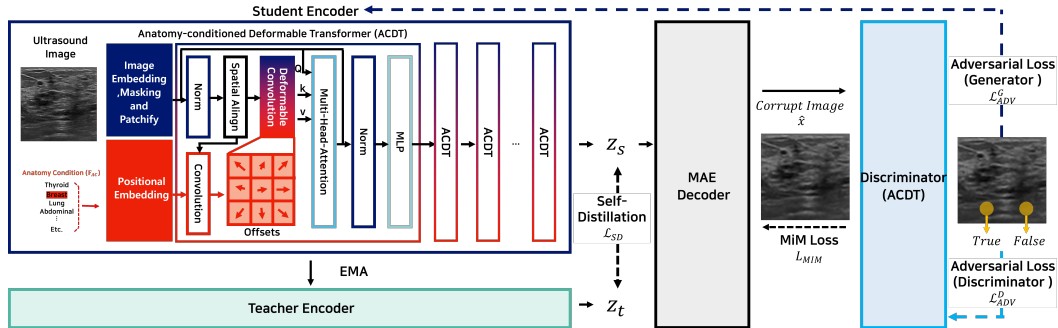

Figure 2: Overall configuration of the proposed A-ViT.

imaging regions, A-ViT introduces an anatomy-specific embedding, ensuring that the model adapts the representation learning process to the targeted anatomical context.

**Anatomy-Conditioned Deformable Transformer (ACDT).** The spatial distribution of diagnostically meaningful features in US images depends strongly on the organ being imaged. For example, Echo typically relies on globally distributed features such as cardiac chambers, whereas breast US emphasizes locally confined features such as breast lesions. To effectively encode these anatomy-dependent spatial relationships among image patches, ACDT employs deformable convolution with offsets conditioned on anatomical context.

Formally, the flattened patch embeddings $x_f \in \mathbb{R}^{N \times C \times P^2}$, are reshaped into 2D spatial patches $x_P \in \mathbb{R}^{N \times C \times P \times P}$, where $C$ denotes the channel dimension and $P \times P$ is the patch resolution. . Sixteen representative anatomical conditions, corresponding to key diagnostic targets in medical US, are encoded as one-hot vectors of dimension $a \in \{0, 1\}^{B \times 16}$. These vectors are projected through a learned embedding layer to produce the anatomical context representation $AC \in \mathbb{R}^{B \times C}$, which is added to the patch tokens as $x_{P,AC} = x_P + AC$.

The $x_{P,AC}$ are then used to determine the kernel offsets $\Delta p_k = g_\theta(x_{P,AC})$. Given kernel weights $w_k$ and $K$ sampling locations, the deformable convolution is defined as

$$y_P(p) = \sum_{k=1}^{K} w_k \, \mathcal{S}(x_P, \, p + \Delta p_k), \tag{1}$$

where $\mathcal{S}(\cdot, \cdot)$ denotes bilinear sampling. Through the mechanism, ACDT adaptively adjusts its receptive field according to the anatomical condition, enabling more accurate and context-sensitive feature extraction for diverse US imaging tasks. Following the deformable convolution, the anatomy-conditioned features $y_P$ are integrated into the transformer block through the multi-head attention mechanism. In this formulation, the anatomy-conditioned output $y_P$ serves as the key and value, while the original patch embeddings $x_P$ are retained as the query:

$$\text{DeformAttn}(x_P, y_P) = \text{Softmax}\left( \frac{(x_P W^Q)(y_P W^K)^\top}{\sqrt{d_k}} \right) (y_P W^V), \tag{2}$$

where $W^Q, W^K, W^V$ are learnable projection matrices. The attention output is subsequently passed through a standard feed-forward network (FFN) with residual connections and normalization, completing the ACDT block. Employing a series of ACDT blocks, the A-ViT is configured, progressively propagating anatomy-conditioned features throughout the network

### 3.4 Learning objective

To optimize the A-ViT for US image representation, we formulate an SSL strategy that integrates multiple complementary objectives. The proposed SSL framework combines masked image modeling (MIM) for local structural recovery, an adversarial objective to preserve high-frequency speckle features, and a self-distillation objective inspired by DINO for global semantic alignment.

**Masked Image Modeling.** Following the scheme of MAE, we randomly mask a subset of input patches and train the model to reconstruct the missing regions. Let $x \in \mathbb{R}^{H \times W}$ denote the input US image, and let $\mathcal{M}$ be the binary mask, where $\mathcal{M}_i = 0$ indicates that the $i$-th patch is masked. Denote the set of masked indices as $\Omega = \{i \mid \mathcal{M}_i = 0\}$. The reconstruction loss is defined as:

$$\mathcal{L}_{\mathrm{MIM}} = \frac{1}{|\Omega|} \sum_{i \in \Omega} \|x_i - \hat{x}_i\|_2^2, \tag{3}$$

where $\hat{x}$ represents the generated image. Unlike NIs, US images contain diagnostically critical high-frequency speckle patterns, which are often blurred when training exclusively with an $\ell_n$-based loss.

**Adversarial loss for high-frequency preservation.** To enhance the model's ability to recognize speckle details, we incorporate an adversarial loss inspired by generative adversarial networks. A discriminator $D(\cdot)$ is trained jointly to distinguish reconstructed patches from real US patches. The adversarial objective is given by:

$$\mathcal{L}_{\mathrm{adv}}^{(D)} = -\mathbb{E}_x[\log D(x)] - \mathbb{E}_{\hat{x}}\big[\log\big(1 - D(\hat{x})\big)\big], \qquad \mathcal{L}_{\mathrm{adv}}^{(G)} = -\mathbb{E}_{\hat{x}}[\log D(\hat{x})]. \tag{4}$$

By combining $\mathcal{L}_{\mathrm{adv}}^{(G)}$ and $\mathcal{L}_{\mathrm{MIM}}$, the model not only reconstructs missing regions but also preserves fine-grained speckle patterns crucial for US interpretation.

**Self-distillation (SD).** While $\mathcal{L}_{\mathrm{adv}}^{(G)}$ and $\mathcal{L}_{\mathrm{MIM}}$ objectives effectively capture local structural cues through patch-level reconstruction, recent studies have demonstrated their limitations in learning discriminative global semantics. To address this, we integrate the DINO framework (Siméoni et al. (2025)), which introduces a self-distillation scheme across augmented views, thereby enforcing cross-view consistency and enhancing global semantic representation beyond local reconstruction. Let $z_s$ and $z_t$ denote the probability distributions from the student and teacher network output feature, respectively. The SD loss is defined as:

$$\mathcal{L}_{\mathrm{SD}} = -\sum_{i=1}^{N} z_t^{(i)} \log z_s^{(i)}. \tag{5}$$

where N is the number of output tokens. This objective encourages the student network to align its predictions with the teacher distribution across diverse augmentations, thereby enhancing the global contextual representation of the model.

**Final objective and adaptive balancing.** The overall self-supervised loss is denoted as

$$\mathcal{L} = \mathcal{L}_{\mathrm{SD}} + \big(\mathcal{L}_{\mathrm{MIM}} + \lambda\, \mathcal{L}_{\mathrm{adv}}^{(G)}\big), \quad \lambda = \frac{\|\nabla \mathcal{L}_{\mathrm{MIM}}\|}{\|\nabla \mathcal{L}_{\mathrm{adv}}^{(G)}\| + \varepsilon}. \tag{6}$$

The parameter $\lambda$ adaptively balances the contribution of $\mathcal{L}_{\mathrm{adv}}^{(G)}$ and $\mathcal{L}_{\mathrm{MIM}}$, based on the gradient magnitude.

## 4 EXPERIMENT

### 4.1 DOWNSTREAM DATASET

To evaluate the applicability of the proposed A-ViT model across a broad spectrum of medical US image recognition tasks, we conducted experiments on five distinct classification tasks and a semantic segmentation task where US imaging serves as a primary diagnostic modality. Table 1 provides an overview of the experimental configurations. Using four publicly available datasets, we assessed the efficacy of the proposed A-ViT model for breast cancer, gallbladder tumors, COVID-19 diagnosis, and Echo left ventricle (LV) segmentation. Additionally, for thyroid cancer and cardiac view classification, we constructed relatively large labeled datasets and conducted the corresponding experiments.

Table 1: Overview of the downstream datasets.

| Task | Disease | Task | size | Public |
|------|---------|------|------|--------|
| Breast (Al-Dhabyani et al. (2020)) | Cancer | Classification | 655 | Yes |
| Gallbladder (Basu et al. (2022)) | Tumor | Classification | 1,255 | Yes |
| Thyroid | Cancer | Classification | 3,428 | No |
| Cardiac | View Cls. | Classification | 39,570 | No |
| Cardiac | Echocardiography | Segmentation | 10,030 | Yes |
| Lung (Born et al. (2021)) | COVID-19 | Classification | 12,108 | Yes |

For breast cancer classification (Cls) we employ the publicly available Breast Ultrasound Images (BUSI) dataset, consisting of 445 malignant and 210 benign tumor cases. The dataset is then split into training (n=505) and testing (n=150) sets for evaluation. For gallbladder tumor identification, we use the public GBCU dataset, which comprises 823 tumors and 432 normal US images collected from 147 patients. The EchoNet-Dynamic dataset, consists of Echo image and the corresponding LV segmentation masks, is used to assess A-ViT on semantic segmentation (Seg) downstream task. The EchoNet-Dynamic dataset is composed of 8,000 train and 2,030 test dataset. To evaluate the performance of A-ViT on lung ultrasound, we utilized an open-access COVID-19 lung US dataset. Lung US videos are collected subject to normal and COVID-19 patients employing low-image quality portable ultrasound devices. The dataset is composed of 9,251 train sets and 2,857 test cases.

In addition, we propose two medical US datasets. First, a thyroid cancer dataset consisting of 3,428 cases of benign (n=1,496) and malignant B-mode images (n=1,932) is annotated under expert radiological and pathological validation. Second, an Echo dataset comprising 15 standard cardiac views is collected and annotated by certified medical professionals. Detailed statistics and annotation protocols are provided in Appendix A.2.

## 4.2 DOWNSTREAM EVALUATION DETAILS

To evaluate performance on downstream tasks, we follow standard practices in SSL (Caron et al. (2021); He et al. (2022)). Specifically, two representative methods, (1) training a linear classifier on frozen backbone features (linear probing), and (2) fine-tuning the backbone on the downstream tasks (fine-tuning), are employed. For comparative analysis, the proposed method is benchmarked against SoTA SSL models in computer vision, MAE (He et al. (2022)), MoCo v3 (He et al. (2020)), iBOT(Zhou et al. (2021)), SigLIP2(Tschannen et al. (2025)) and DINO v3(Siméoni et al. (2025))), medical US, DMAE (Kang et al. (2023b)) and USFM (Jiao et al. (2024)) and multi-modal medical image LVM-Med (MH Nguyen et al. (2023)) domains. Comparative SSL methods employ the ViT-B (Dosovitskiy (2020) with a patch size of 16 as the backbone. A-ViT is configured with the same depth, hidden dimension, and number of attention heads as baselines, thereby maintaining a comparable computational cost across models.

For quantitative assessments, we report AUROC, sensitivity, specificity, and accuracy for breast cancer classification, AUROC and Top-1 accuracy for thyroid, gallbladder, and COVID-19 classification, Dice and mIoU for echocardiography segmentation, and Top-1/Top-3 accuracy for cardiac view classification

## 5 EXPERIMENTAL RESULTS

### 5.1 BREAST CANCER

Table 2 presents the proposed A-ViT in comparison with SSL baselines and a randomly initialized reference. Overall, A-ViT achieves consistently superior performance under both linear probing and fine-tuning, underscoring its ability to learn discriminative representations for breast cancer classification. Under linear probing, A-ViT attains an accuracy of 86.62%, surpassing SSL baselines. By contrast, MAE achieves 77.64% accuracy, reflecting the limitation of NI foundation models in US applications, where capturing fine-grained, high-frequency speckle patterns is essential. With fine-tuning, A-ViT further improves to 93.66%, outperforming the vision-language model SigLIP2 (89.34%) as well as the US-specific foundation model USFM (88.73%).

Table 2: Quantitative assessment of breast cancer classification under linear probing and fine-tuning.

| Method | Linear Probing | | | | Fine-tuning | | | |
|---|---|---|---|---|---|---|---|---|
| | Accuracy | AUROC [CI] | Sensitivity | Specificity | Accuracy | AUROC [CI] | Sensitivity | Specificity |
| Supervised | – | – | – | – | 80.98 | 0.8653 (0.79, 0.92) | 0.6667 | 0.9010 |
| MAE | 77.64 | 0.8211 (0.75, 0.89) | 0.5747 | 0.9000 | 83.09 | 0.8635 (0.80, 0.92) | 0.7636 | 0.8778 |
| MoCo v3 | 83.09 | 0.9143 (0.86, 0.96) | 0.6875 | 0.9239 | 85.91 | 0.9065 (0.85, 0.95) | 0.6897 | **0.9674** |
| iBOT | 84.50 | 0.9090 (0.86, 0.95) | 0.7632 | 0.9023 | 88.02 | 0.9256 (0.88, 0.96) | 0.8347 | 0.9140 |
| SigLIP2 | 77.64 | 0.8396 (0.77, 0.90) | 0.6829 | 0.8352 | 89.34 | 0.9351 (0.89, 0.97) | 0.8200 | 0.9468 |
| Dino v3 | 78.16 | 0.8382 (0.77, 0.90) | 0.6818 | 0.8446 | 84.50 | 0.9172 (0.87, 0.96) | 0.6667 | 0.9326 |
| LVM-Med | 82.39 | 0.8694 (0.81, 0.92) | 0.6304 | **0.9462** | 84.50 | 0.9083 (0.86, 0.95) | 0.8000 | 0.8790 |
| DMAE | 78.16 | 0.8306 (0.75, 0.90) | 0.7188 | 0.8118 | 86.61 | 0.9308 (0.89, 0.97) | 0.7238 | 0.9565 |
| USFM | 82.39 | 0.8927 (0.84, 0.94) | 0.7091 | 0.9022 | 88.73 | 0.9376 (0.89, 0.97) | 0.8163 | 0.9344 |
| **Proposed** | **86.62** | **0.9151 (0.86, 0.96)** | **0.8333** | 0.8917 | **93.66** | **0.9742 (0.95, 0.99)** | **0.9455** | 0.9355 |

Table 3: Ablation study on the effect of loss functions and ACDT in breast cancer classification.

| $\mathcal{L}_{SD}$ | $\mathcal{L}_{MIM}$ | $\mathcal{L}_{adv}$ | ACDT | Dataset | Accuracy | AUROC [CI] | Sensitivity | Specificity | Params |
|---|---|---|---|---|---|---|---|---|---|
| | ✓ | | | Natural Image | 83.09 | 0.8635 (0.80,0.92) | 0.7636 | 0.8778 | 86M |
| | ✓ | | | Ultrasound | 89.43 +6.34 | 0.9380 (0.89,0.97) | 0.8750 | 0.9140 | 86M |
| | ✓ | | ✓ | Ultrasound | 92.25 +2.82 | 0.9664 (0.93,0.99) | 0.8727 | **0.9570** | 95M |
| | ✓ | ✓ | ✓ | Ultrasound | 92.95 +0.70 | 0.9688 (0.94,0.99) | 0.8820 | 0.9464 | 95M |
| ✓ | ✓ | ✓ | ✓ | Ultrasound | **93.66** +0.71 | **0.9742** (0.95,0.99) | **0.9455** | 0.9355 | 95M |

**Ablation Study.** In this section, we analyze the ablation results presented in Table 3, which disentangle the contributions of self-supervision objectives and ACDT. The comparison between natural-image pretraining and US-based pretraining under the MIM objective demonstrates the critical role of domain alignment. While natural-image pretraining yields an accuracy of 83.09%, US-specific pretraining improves performance substantially to 89.43%, suggesting that NI-based representations are insufficient to capture the discriminative cues present in ultrasound. The integration of ACDT leads to 2.79% accuracy improvement, establishing the value of anatomy-guided deformable attention in refining feature representations. In breast cancer classification, the ability to recognize calcifications within breast tumors is particularly important, as these high-frequency patterns often serve as critical diagnostic cues. The incorporation of adversarial loss facilitates representation learning of such high-frequency components, thereby contributing to the observed improvement in classification accuracy. The incorporation of self-distillation loss further enhances performance, suggesting that the learning objective reinforces global feature learning.

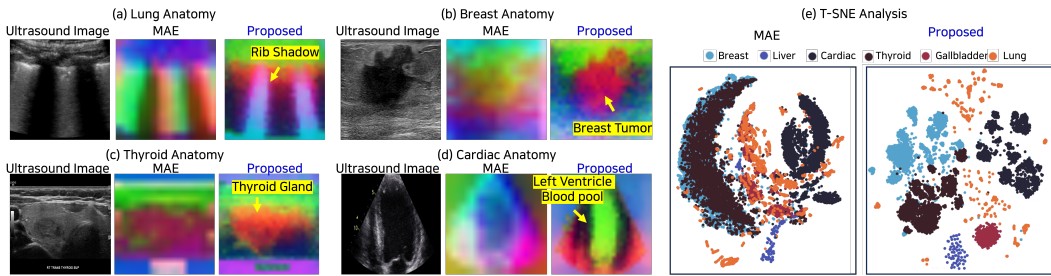

Figure 3: Feature principal component analysis (PCA) and t-SNE analysis of MAE and A-ViT

**Feature PCA and t-SNE Analysis** Fig. 3 demonstrates a principal component analysis (PCA) of features extracted by SSL models, comparing the MAE and the A-ViT across multiple US domains. The visualization qualitatively confirms that the proposed A-ViT yields more discriminative and anatomically faithful feature representations. In the breast domain, A-ViT more precisely delineates lesion morphology, enabling clearer representation of shape-related cues essential for malignancy assessment. In lung imaging, the model effectively captures the pleural line and rib shadow, which are fundamental landmarks for identifying pulmonary abnormalities. For cardiac US, A-ViT provides a distinct representation of the LV region, highlighting its ability to encode clinically relevant cardiac structures. Fig. 3(e) introduces t-SNE analysis of features extracted by MAE and A-ViT across different anatomies. A-ViT yields distinct and separable clusters, thereby demonstrating

Table 4: Quantitative assessments of the networks across multiple US downstream tasks.

| Method | Cardiac Seg. | | Cardiac view Cls. | | Thyroid cancer Cls. | | COVID-19 Cls. | | Gallbladder Tumor Cls. | |
|---|---|---|---|---|---|---|---|---|---|---|
| | Dice | mIoU | Top-1 | Top-3 | Acc. | AUROC [CI] | Acc. | AUROC [CI] | Acc. | AUROC [CI] |
| MAE | 89.21 | 81.06 | 89.07 | 99.09 | 82.50 | 0.9110 (0.89, 0.93) | 80.74 | 0.9346 (0.92, 0.95) | 83.39 | 0.9105 (0.88, 0.94) |
| MoCo v3 | 89.80 | 82.01 | 91.08 | 99.02 | 83.50 | 0.9112 (0.89, 0.93) | 82.22 | 0.9286 (0.92, 0.94) | 84.47 | 0.9237 (0.89, 0.95) |
| SigLIP2 | 90.60 | 83.23 | 90.97 | 99.11 | 85.69 | 0.9330 (0.91, 0.95) | 78.05 | 0.8897 (0.88, 0.90) | 84.11 | 0.9123 (0.88, 0.94) |
| Dino v3 | 90.76 | 83.35 | 89.93 | 99.12 | 86.24 | 0.9428 (0.93, 0.96) | 86.94 | 0.9465 (0.93, 0.96) | 84.84 | 0.9189 (0.89, 0.95) |
| iBOT | 89.45 | 81.37 | 90.59 | 99.13 | 83.68 | 0.9361 (0.92, 0.95) | 81.55 | 0.9494 (0.94, 0.96) | 85.19 | 0.9213 (0.89, 0.95) |
| LVM-Med | 89.41 | 81.30 | 88.64 | 98.72 | 83.42 | 0.9101 (0.89, 0.93) | 85.40 | 0.9359 (0.92, 0.95) | 80.15 | 0.8700 (0.82, 0.91) |
| DMAE | 90.44 | 83.02 | 90.50 | 99.20 | 84.72 | 0.9210 (0.90, 0.94) | 85.99 | 0.9059 (0.89, 0.92) | 84.83 | 0.9019 (0.87, 0.94) |
| USFM | 91.13 | 84.15 | 89.95 | 98.97 | 85.50 | 0.9301 (0.91, 0.95) | 87.67 | 0.9475 (0.94, 0.96) | 86.64 | 0.9347 (0.90, 0.96) |
| **Proposed** | **92.16** | **85.67** | **91.80** | **99.22** | **87.07** | **0.9475 (0.93, 0.96)** | **91.44** | **0.9714 (0.97, 0.98)** | **89.89** | **0.9511 (0.93, 0.97)** |

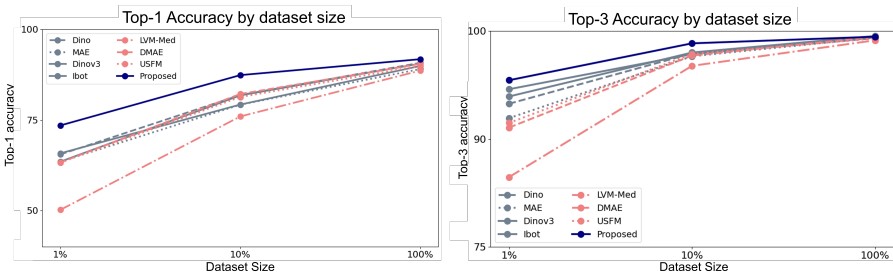

Figure 4: Cardiac view classification by the number of dataset. Top-1 and Top-3 accuracy are presented.

enhanced anatomical discriminability. The results demonstrate that the A-ViT captures anatomy-specific US features with higher fidelity, thereby contributing to stronger generalizability and improved accuracy in downstream diagnostic tasks.

## 5.2 CARDIAC SEGMENTATION

The effectiveness of the proposed A-ViT model is evaluated on the Echo semantic segmentation to demonstrate its capability in dense prediction tasks. A-ViT is evaluated using Dice score and mean intersection over union (mIoU) metrics. To adapt the ViT for dense prediction, a decoder layer of UPerNet (Xiao et al. (2018)) is employed and fine-tune the entire network. Quantitative results, summarized in Table 4, indicate that A-ViT outperforms the baseline models, achieving enhanced Dice score and mIoU, thus demonstrating efficacy in dense prediction scenarios.

## 5.3 CARDIAC VIEW CLASSIFICATION

We perform a multi-class classification experiment across 15 standard Echo views. Given the relatively low inter-view variation in Echo images compared to NIs, Echo video classification requires a more detailed level of feature extraction to capture the precise distinctions between views. Such differences demand highly refined learned representations. Under fine-tuning evaluation, the proposed A-ViT achieves a Top-1 accuracy of 91.80%, demonstrating a substantial improvement over competing SSL baselines. The result emphasizes A-ViT's ability to learn detailed, anatomically relevant features that are crucial for distinguishing between Echo views.

**Ablation Study: effect of dataset size.** To investigate the impact of training dataset size on cardiac view classification, we progressively decreased the dataset size down to 1% of its original volume. As shown in Fig 4, the A-ViT demonstrates less performance degradation than the baseline approaches under such limited data conditions. A comparison of top-1 accuracy reveals that even when trained on only 1% of the data, approximately 0.4K samples, the proposed model outperforms the best-performing baseline by a significant margin. The experiment demonstrates the robustness and efficiency of A-ViT in learning high-quality representations from substantially smaller datasets.

## 5.4 EXTENDED EVALUATION OF MEDICAL ULTRASOUND IMAGE ANALYSIS

**Thyroid cancer classification.** Table.4 presents the performance of the finetuned models in benign and malignant thyroid cancer classification. The proposed method shows the highest accuracy (87.07%) and AUROC (0.9475), demonstrating potential for clinical application of cancerous thyroid tumor diagnosis.

**COVID-19 identification.** The A-ViT is evaluated for COVID-19 identification, a critical diagnostic application in medical US. The method achieves a top-1 accuracy of 91.44%, outperforming comparative approaches. The experiments show the efficacy of the A-ViT in utilizing lung-specific features, demonstrating potential for clinical application of COVID-19 identification through US.

**Gallbladder tumor identification.** A comprehensive experiment is conducted to assess the performance of SSL methods in gallbladder tumor classification. Gallbladder US images are characterized by their convex structure, which presents attributes markedly distinct from those of NIs. The proposed method demonstrates superior performance, achieving the highest accuracy (89.89%) and AUROC (0.9511). The evaluation demonstrates that the A-ViT exhibits robust feature discrimination for gallbladder tumor classification.

## 6 CONCLUSION

In this paper, we introduce Anatomy-aware Representation Learning, a novel framework designed to address the unique challenges of medical US imaging, such as speckle noise, low color variability, and anatomy-specific attributes. By incorporating the A-ViT and leveraging a newly configured large-scale US dataset, ARL enables generalizable feature extraction adaptive to specific anatomical contexts. Experimental results demonstrate the effectiveness of ARL over state-of-the-art baselines in major medical US applications including breast cancer, thyroid cancer, gallbladder tumor, cardiac view classification, and COVID-19 identification, achieving significant improvements in accuracy and AUROC. The experiments demonstrate the potential of the ARL as a reliable and efficient solution for computer-aided diagnostics, offering the foundation for broader integration of AI-driven methods into clinical workflows.

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

# A APPENDIX. DETAILS OF THE PUBLIC DATASET EMPLOYED FOR THE STUDY

Table 5: Summary of ultrasound datasets used for representation learning and downstream applications.

| Dataset Name | Modality | Anatomy | Usage |
| --- | --- | --- | --- |
| BUS-BRA | Ultrasound | Breast | Representation learning |
| BUS-UCLM | Ultrasound | Breast | Representation learning |
| BUV | Ultrasound | Breast | Representation learning |
| DDTI | Ultrasound | Thyroid | Representation learning |
| Thyroid Ultrasound Cine-clip | Ultrasound | Thyroid | Representation learning |
| EchoNet-LVH | Ultrasound | Echocardiography | Representation learning |
| CAMUS | Ultrasound | Echocardiography | Representation learning |
| HMC-QU | Ultrasound | Echocardiography | Representation learning |
| Fatty liver dataset | Ultrasound | Liver | Representation learning |
| FALLMUD | Ultrasound | MSK | Representation learning |
| Ovarian dataset | Ultrasound | Ovarian | Representation learning |
| BUSI | Ultrasound | Breast | Downstream application |
| EchoNet-Dynamic | Ultrasound | Echocardiography | Downstream application |
| POCUS | Ultrasound | Lung | Downstream application |
| GBCU | Ultrasound | Gallbladder | Downstream application |

In this study, we introduce a novel large-scale medical ultrasound dataset, along with detailed descriptions of the thyroid cancer and cardiac classification datasets proposed for evaluation.

## A.1 LARGE-SCALE MEDICAL ULTRASOUND DATASET

The proposed large-scale medical ultrasound dataset comprises 5.2 million images, integrating data from 11 publicly available ultrasound datasets (Gómez-Flores et al. (2024); Lin et al. (2022); Vallez et al. (2025); Pedraza et al. (2015b); AIMI; Ouyang et al. (2019); Duffy et al. (2022); Leclerc et al. (2019); Degerli et al. (2024); Byra et al. (2018); Cunningham et al. (2018); van den Heuvel et al. (2018); Yap et al. (2017)) and additional collections from 12 medical institutions in the United States, 2 in South Korea, and 1 in India. Data acquisition procedures were conducted under Institutional Review Board (IRB) approval and in strict compliance with the Health Insurance Portability and Accountability Act (HIPAA) and the General Data Protection Regulation (GDPR), thereby ensuring both ethical and legal data handling. The dataset encompasses 16 distinct categories of medical ultrasound imaging. For publicly available datasets, anatomical classes are explicitly defined at the dataset level, and for newly curated data, anatomical labels are assigned using exam types and metadata. The anatomical classes consist of Abdominal, Aorta, Bladder, Breast, Cardiac, Doppler, Esophageal, Gallbladder, Kidney, Liver, Lung, Muscle, Oncology, Soft tissue, Testicle, Thyroid, and Other, thereby comprehensively representing diverse anatomical regions and clinical applications. To the best of our knowledge, this constitutes the largest publicly reported medical ultrasound dataset to date, exceeding existing datasets by an order of magnitude in size. A detailed summary of the datasets incorporated in this large-scale collection is provided in the table.

## A.2 THYROID ULTRASOUND TUMOR DATASET

The thyroid cancer evaluation dataset is also presented in this study, specifically configured for evaluating classification performance. The thyroid dataset includes a total of 3,428 thyroid tumor B-mode ultrasound images, each annotated for benign (n=1,496) or malignant (=1,932) status. The annotations are conducted with three certified sonographers, with an expert radiologist confirming the annotations to establish ground truth. Ground truth labeling for malignant tumors is determined based on pathological results from tissue biopsies confirming malignancy. For benign tumors, cases are labeled either through pathology-confirmed benignity or through follow-up examinations over a period of at least 2 years, during which consistent radiological findings indicated benign features. The thyroid cancer dataset is divided into training and testing subsets, with 2,662 images used for

training and 766 images reserved for testing. The proposed dataset is, to the best of our knowledge, the largest publicly reported thyroid ultrasound dataset available to date

### A.3 ECHOCARDIOGRAPHY 15 VIEW DATASET

The echocardiography classification evaluation dataset comprises 39,570 echocardiography images collected for classification tasks. The images are annotated by three experienced sonographers with extensive echocardiography expertise. The dataset includes standard echocardiography views classified into 15 categories: Apical 2-Chamber (A2C), Apical 3-Chamber (A3C), Apical 4-Chamber (A4C), Parasternal Long-Axis (PLAX), Parasternal Short-Axis at the Apex (PSAX_APEX), Parasternal Short-Axis at the Aortic Valve Level (PSAX_AV), Parasternal Short-Axis at the Mitral Valve Level (PSAX_MV), Parasternal Short-Axis at the Papillary Muscle Level (PSAX_PM), Right Ventricular Apical 4-Chamber (RVA4C), Right Ventricular Inflow View (RVIV), Right Ventricular Outflow View (RVOV), Subcostal 4-Chamber View (SC4V), Subcostal Long-Axis View (SCLA), Suprasternal Notch View (SSN). The dataset is split into training and testing subsets, with 33,520 images used for training and 6,050 images reserved for testing.

## B APPENDIX. IMPLEMENTATION DETAILS

**Linear probing.** For linear probing on breast cancer classification, we employed a classification head of a linear classifier processing the global average pooled output feature of vision transformer. The vision transformer encoder weight is frozen, and weight of the classifier head is optimized through stochastic gradient descent with a momentum of 0.9. During training, we employed horizontal flip and random resize crop augmentation with image height and width of 224 and 224, respectively.

**Fine-tuning.** The fine-tuning downstream evaluation is conducted for breast, thyroid, and gallbladder cancer classification, COVID-19 identification, and cardiac view classification. In contrast to linear probing, which trains only a shallow classification head on frozen features, fine-tuning optimizes every parameter of the Vision Transformer encoder together with the classification head. AdamW optimizer is employed with a base learning rate of 0.001, with a cosine decay schedule with a 5-epoch warm-up, $\beta_1 = 0.9$, $\beta_2 = 0.999$.

**Semantic segmentation.** Echocardiography left ventricle blood-pool segmentation is experimented for the assessment of the anatomy-aware vision transformer for dense prediction tasks. UPerNet (Xiao et al. (2018)) configuration is employed. The network is trained using the AdamW optimizer with a learning rate of 0.0001, with a learning objective of minimizing the unweighted sum between cross-entropy loss and Dice loss as proposed in Ma et al. (2024).

## C APPENDIX. EFFECT OF PRETRAINING DATASET SCALE

To assess the impact of pretraining dataset scale on representation learning quality, we varied the number of ultrasound images used for self-supervised pretraining from 5.2M down to 52K and evaluated A-ViT on breast cancer classification (fine-tuning). The results in Table 6 show that both accuracy and AUROC degrade as the dataset size decreases, highlighting the importance of large-scale and diverse ultrasound data for effective and generalizable representation learning.

Table 6: Effect of pretraining dataset scale on breast cancer classification (fine-tuning).

| Pretraining Dataset Size | Accuracy | AUROC |
|---|---|---|
| 5.2M (100%) | 93.66 | 0.9742 |
| 520K (10%) | 90.84 | 0.9581 |
| 52K (1%) | 87.23 | 0.9353 |

## D APPENDIX. NATURAL- AND ULTRASOUND-IMAGE PRETRAINING

We further compared self-supervised methods pretrained on natural-image datasets and on the proposed ultrasound dataset. Table 7 reports breast cancer classification results (fine-tuning). Across

MAE, DINO, and iBOT, ultrasound-specific pretraining consistently improves accuracy and AU-ROC over ImageNet pretraining, confirming the benefit of modality-aligned pretraining. The proposed A-ViT, trained on ultrasound data with ACDT, achieves the best overall performance.

Table 7: Breast cancer classification performance under natural-image vs ultrasound-image pretraining.

| Method | Pretraining Data | Accuracy | AUROC [CI] | Sensitivity | Specificity |
|--------|------------------|----------|------------|-------------|-------------|
| MAE | Natural Image Dataset | 83.09 | 0.8635 (0.8038, 0.9193) | 0.76 | 0.88 |
| MAE | Proposed Ultrasound Dataset | 89.43 | 0.9380 (0.8915, 0.9724) | 0.88 | 0.91 |
| DINO | Natural Image Dataset | 84.50 | 0.9172 (0.8652, 0.9605) | 0.67 | 0.93 |
| DINO | Proposed Ultrasound Dataset | 90.14 | 0.9682 (0.9430, 0.9900) | 0.89 | 0.91 |
| iBOT | Natural Image Dataset | 88.02 | 0.9256 (0.8805, 0.9644) | 0.83 | 0.91 |
| iBOT | Proposed Ultrasound Dataset | 90.85 | 0.9662 (0.9400, 0.9859) | 0.87 | 0.93 |
| A-ViT | Proposed Ultrasound Dataset | **93.66** | **0.9742 (0.9481, 0.9930)** | **0.95** | **0.94** |

## E    APPENDIX. COMPARISON OF CONDITIONING MECHANISMS

In order to assess the proposed ACDT, we compared alternative conditioning mechanisms for incorporating anatomy information into the transformer backbone, including cross-attention, FiLM, and LoRA. Table 8 summarizes breast cancer classification performance (fine-tuning). The proposed ACDT achieves the highest accuracy and AUROC, indicating that anatomy-conditioned deformable attention is particularly effective for capturing anatomy-specific feature distributions in ultrasound.

Table 8: Comparison of conditioning mechanisms on breast cancer classification (fine-tuning).

| Conditioning Scheme | Accuracy | AUROC |
|---------------------|----------|-------|
| Cross-attention | 90.84 | 0.9578 |
| FiLM | 90.14 | 0.9624 |
| LoRA | 92.25 | 0.9692 |
| Proposed ACDT | **93.66** | **0.9742** |

## F    APPENDIX. ADAPTIVE LOSS WEIGHTING

We evaluated the impact of the adaptive loss-weighting scheme that balances the masked-image modeling and adversarial objectives. MSE-based losses predominantly capture low-frequency structure, whereas adversarial losses better preserve high-frequency content. Fixed weighting can lead to imbalanced optimization of the complementary image features. Adaptive weighting prevents domination by a single loss term and stabilizes representation learning. Table 9 presents breast cancer classification performance under constant and adaptive weighting. Adaptive weighting yields higher accuracy and AUROC, supporting its effectiveness in stabilizing optimization and jointly modeling low- and high-frequency image characteristics.

Table 9: Effect of loss weighting strategy on breast cancer classification (fine-tuning).

| Loss Weighting Strategy | Accuracy | AUROC |
|-------------------------|----------|-------|
| Constant weighting | 92.95 | 0.9701 |
| Adaptive weighting | **93.66** | **0.9742** |

## G    APPENDIX. MASKED-IMAGE RECONSTRUCTION ANALYSIS

To assess how well the learned representations capture ultrasound-specific characteristics, we conducted a masked-image reconstruction experiment. Masked B-mode images were encoded using (i) a standard ViT pretrained as a MAE on natural images and (ii) the proposed A-ViT encoder, followed by a shared MAE decoder.

Figure 5 presents side-by-side reconstructions for both encoders under identical masking patterns. Reconstructions obtained from A-ViT more faithfully preserve high-frequency speckle patterns and fine-grained structural details, whereas the natural-image-pretrained ViT produces overly smoothed outputs.

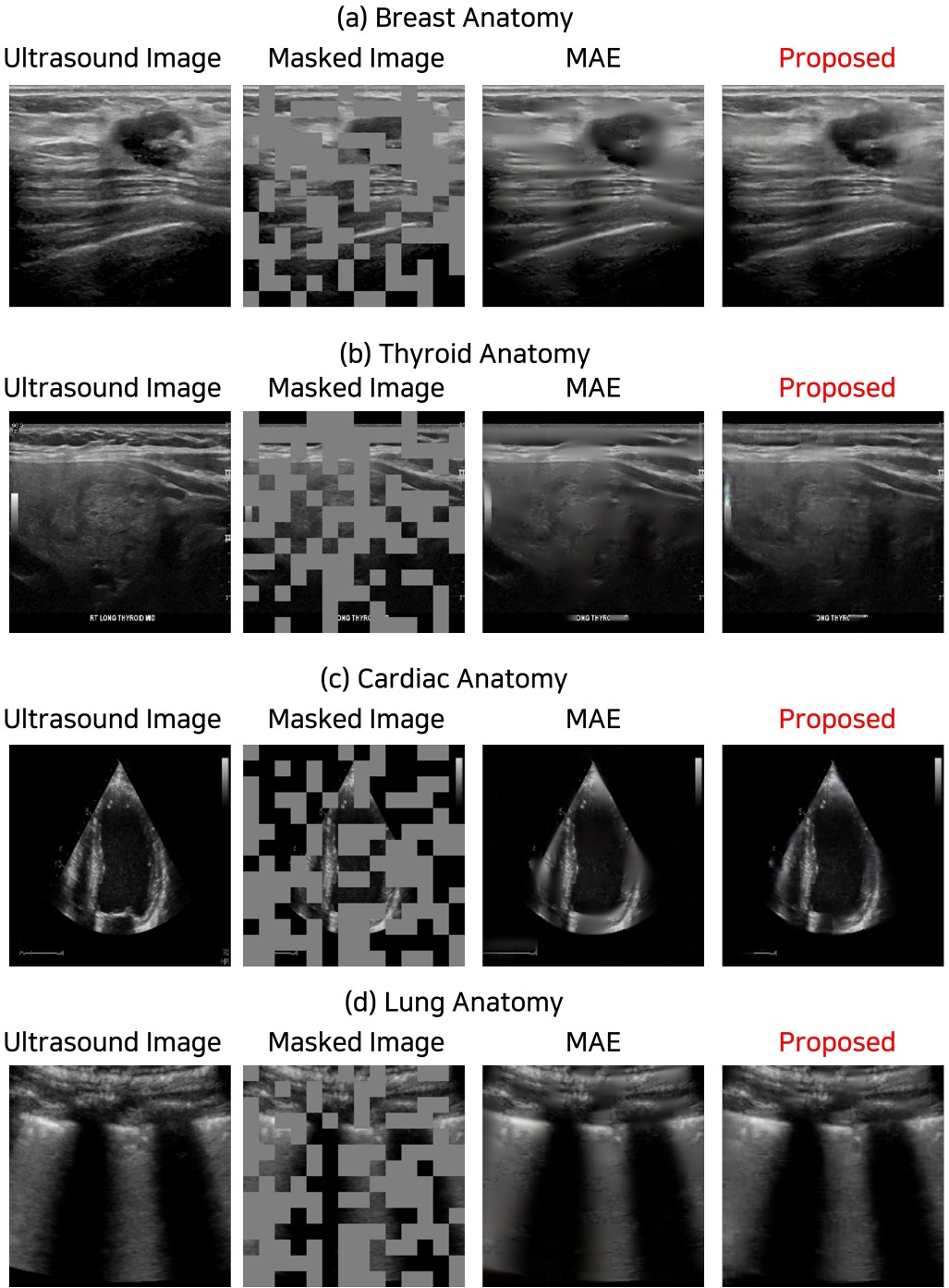

Figure 5: Masked-image reconstruction comparison between a standard MAE and the proposed A-ViT with a MAE decoder. A-ViT reconstructions preserve ultrasound-specific speckle patterns and fine structural details.

## H    APPENDIX. COMPUTATIONAL COST AND INFERENCE SPEED

To evaluate the clinical deployability of A-ViT, we analyze its computational complexity and inference speed in comparison with a standard Vision Transformer (ViT-B/16) backbone. FLOPs are computed for a single $224 \times 224$ B-mode ultrasound image, and inference speed is measured on a single NVIDIA RTX 4090 GPU.

As summarized in Table 10, A-ViT introduces only a modest increase in parameter count and FLOPs relative to the standard ViT, while providing substantially improved anatomy-aware representations in downstream tasks.

Table 10: Computational cost and inference speed comparison between a standard Vision Transformer and the proposed A-ViT.

| Model | Parameters | FLOPs | Inference Speed (ms / image) |
|---|---|---|---|
| Vision Transformer (ViT-B/16) | 86M | 17.58G | 12.1 |
| A-ViT (proposed) | 95M | 17.66G | 16.6 |

## I    APPENDIX. ETHICS STATEMENT

All ultrasound data utilized in this study were fully anonymized in compliance with the Health Insurance Portability and Accountability Act (HIPAA) and the General Data Protection Regulation (GDPR). The studies were conducted under the institutional review board approval of Seoul National University Bundang Hospital (SNUBH).

