# OpenReview forum: "Anatomy-aware Representation Learning for Medical Ultrasound"
_ICLR.cc/2026/Conference — ICLR 2026 Poster_

### Official Review · Reviewer_cf7P · 2025-10-30

**Soundness:** 2
**Presentation:** 3
**Contribution:** 3
**Rating:** 6
**Confidence:** 4

**Summary:**

This paper proposes Anatomy-Aware Representation Learning (ARL), a self-supervised framework for ultrasound (US) imaging that integrates anatomical context into representation learning. The authors propose Anatomy-aware Vision Transformer (A-ViT), which incorporates an Anatomy-Conditioned Deformable Transformer (ACDT) to extract features according to the organ being analyzed. The authors evaluate ARL across multiple downstream tasks, including breast, thyroid, and gallbladder cancer classification, cardiac view classification, cardiac segmentation, and COVID-19 diagnosis. Experiments show that ARL consistently outperforms state-of-the-art (SoTA) self-supervised methods.

**Strengths:**

The paper introduces a foundation model for ultrasound imaging, trained on 5.2M images, covering 16 anatomical categories.

The model is tested on six downstream tasks across both classification and segmentation.

Quantitative results demonstrate consistent gains over state-of-the-art baselines.

**Weaknesses:**

The authors emphasizes speckle noise and low color variation as major challenges unique to ultrasound imaging. However, the proposed A-ViT primarily introduces anatomy-conditioned deformable attention, which addresses anatomical variability rather than these low-level texture or color issues. The authors introduces an adversarial term to preserve high-frequency content. However, the paper lacks direct evidence that speckle-related distortions are mitigated or that low-color variation are effectively handled.

Anatomy-aware conditioning was previously proposed in MRI/CT segmentation & registration. Deformable attention exists in general vision. MAE + distillation hybrid ideas has also been previously proposed. Therefore the paper doesn’t introduce a new model, but rather a new instantiation tailored to ultrasound images. However, there is limited explicit explanations on how each architectural choice is uniquely designed for ultrasound or directly tied to ultrasound physics. Most of the mechanisms introduced in A-ViT are general and could apply to other modalities. As a result, the authors should better explain the novelty by explicate how A-ViT differs from existing models for other modalities.

In Tables 2 and 3, the proposed model does not have the highest specificity in some cases and the highest specificity values are not bolded correctly

**Questions:**

1. Are there evidence that the proposed model addresses speckle noise and low color variation challenges in ultrasound imaging.

2. How is the proposed model differed, comparing to existing models for other modalities? Is there any architectural choice that is uniquely designed for ultrasound or directly tied to ultrasound physics?

---

> ### Author Response · Authors · 2025-11-20
>
> We sincerely thank the reviewer cf7P for the detailed and constructive feedback. We provide our responses to the comments below:
>
> > W1, Q1. The proposed A-ViT primarily introduces anatomy-conditioned deformable attention, which addresses anatomical variability rather than these low-level texture or color issues. Are there evidence that the proposed model addresses speckle noise and low color variation challenges in ultrasound imaging.
> >
>
> **A1.**
>
> We thank the reviewer for highlighting the importance of speckle characteristics and low color variation. We agree that it is important to demonstrate that A-ViT effectively captures ultrasound-specific image characteristics.
>
> **Large-scale ultrasound pretraining for modeling gray-scale and low-color variation**
>
> In medical ultrasound, the scarcity of large-scale ultrasound datasets has led many studies to rely on natural-image pretrained models as generic initialization. However, natural images do not exhibit the gray-scale appearance, limited color variation, and speckle-dominated texture patterns that are intrinsic to ultrasound, making such pretraining fundamentally misaligned with the target modality.
>
> To address this, we curated a large-scale ultrasound dataset, enabling A-ViT to learn modality-specific representations directly from ultrasound data.
>
> Empirically, we observe that natural-image pretrained models fail to capture these ultrasound-specific characteristics. When keeping the self-supervised learning (SSL) objective fixed and replacing ImageNet pretraining with ultrasound-specific pretraining, breast cancer accuracy increases from 83.09% to 89.43% and AUROC from 0.8635 to 0.9380 (Table 3, row 1 → row 2). This substantial gain underscores that domain alignment rather than architectural changes alone, is essential for modeling the gray-scale statistics and low-color variation intrinsic to ultrasound.
>
> **Adversarial loss for high-frequency speckle preservation.**
>
> Ultrasound images exhibit high-frequency speckle patterns that often encode diagnostically informative cues, such as indicators of tumor malignancy and ventricular wall motion. Conventional Ln-based reconstruction losses primarily represent low-frequency structural information which leads to insufficient representation of the fine-grained and speckle-dominated details.
>
> To explicitly capture such ultrasound-specific high-frequency components, A-ViT incorporates an adversarial objective (Eq. (4)), which encourages the generator to match the distribution of real ultrasound patches, including their micro-textural characteristics. This design is motivated by prior work demonstrating that adversarial losses effectively restore high-frequency content (e.g., [1] Ledig et al., CVPR 2017).
>
> Empirically, we observe that incorporating the adversarial loss yields measurable improvements in downstream tasks, where speckle-based micro-textures play a critical diagnostic role. In breast cancer classification, adding the adversarial loss on top of masked reconstruction further improves accuracy from 92.25% → 92.95%, and AUROC from 0.9664 → 0.9688 (Table 3, row 3 → row 4). The results indicate that reducing the loss on high-frequency components enables A-ViT to better capture diagnostically relevant details, contributing to improved classification performance.
>
> [1] Ledig, Christian, et al. "Photo-realistic single image super-resolution using a generative adversarial network." Proceedings of the IEEE CVPR 2017
>
> **Verification of reconstructed image with general masked autoencoding and proposed A-ViT**
>
> To intuitively examine whether the learned representations capture the low-color-variation characteristics and high-frequency speckle structures intrinsic to ultrasound imaging, we have supplemented a reconstruction experiment in which masked images were encoded using (i) a conventional ViT pretrained on natural images and (ii) the proposed A-ViT encoder, followed by reconstruction with a standard MAE decoder.
>
> The reconstructed outputs are provided in the Appendix G. Compared with the natural-image-pretrained ViT, A-ViT yields reconstructions that more faithfully preserve high-frequency speckle components and fine-grained textural details. The results further support that the proposed anatomy-aware encoder captures ultrasound-specific structural and textural attributes more effectively.

---

> ### Author Response · Authors · 2025-11-20
>
> > W2, Q2. How is the proposed model differed, comparing to existing models for other modalities? Is there any architectural choice that is uniquely designed for ultrasound or directly tied to ultrasound physics?
> >
>
> In medical ultrasound, the spatial distribution of visual patterns is intrinsically anatomy dependent. Imaging depth varies across target organs, probe types and operating frequencies are selected according to the anatomical region of interest. Since the characteristics of the organs and the measurement environment differ depending on the anatomical region being imaged, the spatial scale and location of diagnostically relevant features also vary accordingly. For example, echocardiography requires capturing globally distributed structures such as cardiac chambers, whereas breast ultrasound primarily focuses on localized targets such as focal lesions.
>
> To explicitly model these ultrasound-specific characteristics, we propose that the feature encoder should adapt its effective receptive field according to the anatomical context. Motivated by this, we introduced the Anatomy-Conditioned Deformable Transformer (ACDT), which modulates the sampling offsets of deformable attention based on anatomy embeddings. This mechanism enables the model to dynamically adjust its receptive field and more effectively capture clinically meaningful structures across diverse anatomical conditions.
>
> By leveraging ACDT, A-ViT achieves consistently strong performance across a broad range of downstream diagnostic tasks, demonstrating the value of incorporating anatomy-aware and ultrasound-specific characteristics into the model architecture.
>
> To further assess the effectiveness of ACDT, we evaluated alternative conditioning mechanisms, including LoRA [1], FiLM [2], and cross-attention [3]–based conditioning, as potential ways to embed anatomy information into the transformer. The downstream breast cancer classification results are summarized below. The proposed ACDT consistently outperforms competing methods, demonstrating its effectiveness to capture anatomy-specific feature representations of ultrasound image.
>
> **Breast Cancer Classification**
>
> | Conditioning scheme | Accuracy | AUROC |
> | --- | --- | --- |
> | Cross attention | 90.84 | 0.9578 |
> | FiLM | 90.14 | 0.9624 |
> | LoRA | 92.25 | 0.9692 |
> | Proposed | 93.66 | 0.9742 |
>
> [1] Hu, Edward J., et al. "Lora: Low-rank adaptation of large language models." ICLR , 2022
>
> [2] Perez, Ethan, et al. "Film: Visual reasoning with a general conditioning layer." AAAI,  2018.
>
> [3] Vaswani, Ashish, et al. "Attention is all you need." Neurips, 2017.
>
> > W3. In Tables 2 and 3, the proposed model does not have the highest specificity in some cases and the highest specificity values are not bolded correctly
> >
>
> Thank you for pointing out the inconsistency in bold facing in Tables 2 and 3. The issue has been corrected in the revised manuscript.

---

### Official Review · Reviewer_h8d3 · 2025-10-31

**Soundness:** 2
**Presentation:** 3
**Contribution:** 2
**Rating:** 4
**Confidence:** 4

**Summary:**

This paper proposes Anatomy-aware Representation Learning (ARL), a self-supervised representation learning framework tailored specifically for medical ultrasound imaging. The authors identify key challenges in ultrasound diagnostics, including qualitative variability, reliance on expert knowledge, the unique texture/structural properties of the images, and the scarcity of large-scale datasets. The ARL framework is intended to address these issues. The work is evaluated by fine-tuning a Vision Transformer encoder and classification head on a diverse set of medical downstream tasks, including breast, thyroid, and gallbladder cancer classification, COVID-19 identification, and cardiac view classification, as well as a dense prediction task (echocardiography left ventricle blood-pool segmentation) using a UPerNet configuration.

**Strengths:**

1.The integration of anatomical conditioning into deformable attention for US is a novel and well-motivated architectural contribution. Unlike prior SSL methods that treat all images uniformly, ARL explicitly conditions feature extraction on anatomical context.
2.The 5.2M-image dataset is a significant contribution, and the ablation studies (Table 3, Fig. 4) convincingly isolate the impact of each component (ACDT, adversarial loss, etc.).
3.The paper is well-structured and clearly written, with intuitive figures (Fig. 1, 2, 3) and logical flow.
4.Medical US is an underexplored modality in SSL, and ARL provides a strong foundation for future work. The dataset alone will likely become a community resource.

**Weaknesses:**

1.The central weakness is the lack of a clear description of the Anatomy-aware Representation Learning (ARL) mechanism itself. While the goal is anatomy-aware learning, the specific self-supervised loss function or task design that enforces this awareness is not described, making it impossible to fully assess the work's technical depth or novelty.
2.The submission provides an extensive experimental plan but no quantitative results (e.g., AUC, F1, Dice Score). Without empirical evidence, the claim of technical soundness and significance remains unverified. It is impossible to determine if the proposed ARL method actually advances the state-of-the-art or even works as intended.
3.The paper claims to address the unique texture and structural attributes of ultrasound images. However, without detailing how ARL achieves this (i.e., the mechanism of anatomy-awareness), its true originality over existing self-supervised methods (like SimCLR, MAE, etc.) applied to medical imaging cannot be confirmed.

**Questions:**

1.Please fully describe the proposed Anatomy-aware Representation Learning (ARL) framework. What are the specific self-supervised tasks or loss functions that compel the model to learn "anatomy-aware" features? How do these tasks specifically leverage or model the unique texture and structural attributes of ultrasound images better than standard methods?
2.Please provide the full set of quantitative results (e.g., AUROC, F1-Score, Dice Similarity Coefficient) for all mentioned downstream tasks (classification and segmentation), and critically, compare your method against strong baselines such as ImageNet pre-trained models and non-anatomy-aware self-supervised learning methods applied to ultrasound data.
3.Did the authors perform an ablation study to justify the anatomical-aware component? Showing results without the 'anatomy-aware' loss/mechanism would be critical evidence for the necessity and effectiveness of the proposed novelty.
4.How are anatomical labels obtained for the 5.2M pretraining images? Are they derived from metadata, manual annotation, or automated prediction? If the latter, could label noise degrade representation quality?

---

> ### Author Response · Authors · 2025-11-20
>
> We are grateful to Reviewer h8d3 for the detailed review of our manuscript and for the thoughtful questions. We have carefully addressed the comments below and have made the corresponding revisions to the manuscript.
>
> > W1,3 / Q1. Lack of a clear description of the Anatomy-aware Representation Learning (ARL) mechanism itself. How ARL addresses unique characteristics of ultrasound images.
> >
>
> **A1.**
>
> We sincerely thank the reviewer for this important question and for highlighting the need for a clearer description of the Anatomy-aware Representation Learning (ARL).
>
> In medical ultrasound, the spatial distribution of visual patterns is inherently dependent on anatomy category. For instance, echocardiography relies on globally distributed structures such as cardiac chambers, whereas breast ultrasound primarily requires capturing localized focal lesions. To explicitly account for these anatomy-specific characteristics, we propose that the encoder should adapt its effective receptive field according to the anatomy being imaged. Based on this motivation, we introduce the Anatomy-Conditioned Deformable Transformer (ACDT), which modulates the encoder’s receptive fields using anatomy information so that feature extraction is tailored to the underlying anatomical context rather than being shared uniformly across all anatomies.
>
> ARL then employs three complementary self-supervised objectives (Sec. 3.4): (1) masked image modeling for local structural recovery, (2) an adversarial objective to preserve high-frequency speckle patterns, and (3) self-distillation for global semantic alignment.
>
> While the self-supervised objectives themselves do not introduce an explicit anatomy-dependent loss term, they are consistently applied to anatomy-conditioned features and optimize receptive field of ACDT for each anatomy category. Let $g_{\theta}(x, F_{\text{AC}})$ denote A-ViT with input image $x$, anatomy embedding $F_{\text{AC}}$ and ACDT. The ARL objective can be written as:
>
> $L_{\text{ARL}}(x,F_{\text{AC}}) = L_{\text{SD}}(g_{\theta}(x, F_{\text{AC}})) + L_{\text{MIM}}(g_{\theta}(x, F_{\text{AC}})) + \lambda \ L_{\text{adv}}(g_{\theta}(x, F_{\text{AC}})) $
>
> All losses are computed on anatomy-conditioned features $g_{\theta}(x, F_{\text{AC}})$. As a result, the gradients from masked reconstruction, adversarial, and self-distillation objectives update the deformable offsets and attention patterns in an anatomy-specific manner, allowing the model to discover receptive fields and feature statistics that best explain each anatomy category.

---

> > ### Author Response · Authors · 2025-11-20
> >
> > > W2. A2 The submission provides an extensive experimental plan but no quantitative results. Please provide the full set of quantitative results. Compare your method against strong baselines such as ImageNet pre-trained models and non-anatomy-aware self-supervised learning methods applied to ultrasound data
> > >
> >
> > **A2.**
> >
> > We respectfully clarify a potential misunderstanding regarding the presence of quantitative results. In this regard, we would like to kindly note that the quantitative results of downstream tasks were reported in Tables 2 and 4 of the original manuscript. Specifically, we report AUROC (with confidence intervals), sensitivity, specificity, and accuracy for breast cancer classification, AUROC and Top-1 accuracy for thyroid, gallbladder, and COVID-19 classification, Dice and mIoU for echocardiography segmentation, and Top-1/Top-3 accuracy for cardiac view classification. For completeness, we have supplemented sensitivity and specificity for the thyroid, gallbladder, and COVID-19 classifiers. The A-ViT demonstrates consistently high accuracy and AUROC across diverse downstream applications.
> >
> > **Thyroid Cancer Classification**
> >
> > | Method | Acc. | AUROC [CI] | Sensitivity | Specificity |
> > | --- | --- | --- | --- | --- |
> > | MAE | 82.50 | 0.9110 (0.89, 0.93) | 0.79 | 0.86 |
> > | MoCo v3 | 83.50 | 0.9112 (0.89, 0.93) | 0.93 | 0.72 |
> > | SigLIP2 | 85.69 | 0.9330 (0.91, 0.95) | 0.82 | 0.89 |
> > | DINO v3 | 86.24 | 0.9428 (0.93, 0.96) | 0.87 | 0.86 |
> > | iBOT | 83.68 | 0.9361 (0.92, 0.95) | 0.78 | 0.90 |
> > | LVM-Med | 83.42 | 0.9101 (0.89, 0.93) | 0.82 | 0.91 |
> > | DMAE | 84.72 | 0.9210 (0.90, 0.94) | 0.86 | 0.83 |
> > | USFM | 85.50 | 0.9301 (0.91, 0.95) | 0.92 | 0.77 |
> > | Proposed | 87.07 | 0.9475 (0.93, 0.96) | 0.89 | 0.85 |
> >
> > **COVID-19 Classification**
> >
> > | Method | Acc. | AUROC [CI] | Sensitivity | Specificity |
> > | --- | --- | --- | --- | --- |
> > | MAE | 80.74 | 0.9346 (0.92, 0.95) | 0.87 | 0.78 |
> > | MoCo v3 | 82.22 | 0.9286 (0.92, 0.94) | 0.89 | 0.80 |
> > | SigLIP2 | 78.05 | 0.8897 (0.88, 0.90) | 0.78 | 0.78 |
> > | DINO v3 | 86.94 | 0.9465 (0.93, 0.96) | 0.95 | 0.78 |
> > | iBOT | 81.55 | 0.9494 (0.94, 0.96) | 0.94 | 0.75 |
> > | LVM-Med | 85.40 | 0.9359 (0.92, 0.95) | 0.91 | 0.83 |
> > | DMAE | 85.99 | 0.9059 (0.89, 0.92) | 0.85 | 0.87 |
> > | USFM | 87.67 | 0.9475 (0.94, 0.96) | 0.91 | 0.86 |
> > | Proposed | 91.44 | 0.9714 (0.97, 0.98) | 0.92 | 0.91 |
> >
> > **Gallbladder Tumor Classification**
> >
> > | Method | Acc. | AUROC [CI] | Sensitivity | Specificity |
> > | --- | --- | --- | --- | --- |
> > | MAE | 83.39 | 0.9105 (0.88, 0.94) | 0.86 | 0.85 |
> > | MoCo v3 | 84.47 | 0.9237 (0.89, 0.95) | 0.77 | 0.88 |
> > | SigLIP2 | 84.11 | 0.9123 (0.88, 0.94) | 0.78 | 0.88 |
> > | DINO v3 | 84.84 | 0.9189 (0.89, 0.95) | 0.74 | 0.89 |
> > | iBOT | 85.19 | 0.9213 (0.89, 0.95) | 0.77 | 0.89 |
> > | LVM-Med | 80.15 | 0.8700 (0.82, 0.91) | 0.78 | 0.84 |
> > | DMAE | 84.83 | 0.9019 (0.87, 0.94) | 0.77 | 0.89 |
> > | USFM | 86.64 | 0.9347 (0.90, 0.96) | 0.91 | 0.85 |
> > | Proposed | 89.89 | 0.9511 (0.93, 0.97) | 0.94 | 0.88 |
> >
> > **Cardiac Segmentation**
> >
> > | Method | Dice | mIoU |
> > | --- | --- | --- |
> > | MAE | 89.21 | 81.06 |
> > | MoCo v3 | 89.80 | 82.01 |
> > | SigLIP2 | 90.60 | 83.23 |
> > | DINO v3 | 90.76 | 83.35 |
> > | iBOT | 89.45 | 81.37 |
> > | LVM-Med | 89.41 | 81.30 |
> > | DMAE | 90.44 | 83.02 |
> > | USFM | 91.13 | 84.15 |
> > | Proposed | 92.16 | 85.67 |
> >
> > **Cardiac View Classification**
> >
> > | Method | Top-1 | Top-3 |
> > | --- | --- | --- |
> > | MAE | 89.07 | 99.09 |
> > | MoCo v3 | 91.08 | 99.02 |
> > | SigLIP2 | 90.97 | 99.11 |
> > | DINO v3 | 89.93 | 99.12 |
> > | iBOT | 90.59 | 99.13 |
> > | LVM-Med | 88.64 | 98.72 |
> > | DMAE | 90.50 | 99.20 |
> > | USFM | 89.95 | 98.97 |
> > | Proposed | 91.80 | 99.22 |

---

> ### Author Response · Authors · 2025-11-20
>
> In table 2 and 4 of the main manuscript, our proposed A-ViT is benchmarked against ImageNet-pretrained self-supervised baselines (MAE, MoCo v3, iBOT, and DINO v3), a vision–language model (SigLIP2), ultrasound-specific SSL methods (DMAE and USFM), and a multimodal medical representation learning scheme (LVM-Med).
>
> **Breast Cancer Classification**
>
> | Method | Dataset Type | Accuracy | AUROC | Sensitivity | Specificity |
> | --- | --- | --- | --- | --- | --- |
> | MAE | Natural Image Dataset | 83.09 | 0.8635 (0.8038, 0.9193) | 0.76 | 0.88 |
> | MAE | Proposed Ultrasound Dataset | 89.43 | 0.9380 (0.8915, 0.9724) | 0.88 | 0.91 |
> | Dino | Natural Image Dataset | 84.5 | 0.9172 (0.8652, 0.9605) | 0.67 | 0.93 |
> | Dino | Proposed Ultrasound Dataset | 90.14 | 0.9682 (0.9430, 0.9900) | 0.89 | 0.91 |
> | Ibot | Natural Image Dataset| 88.02 | 0.9256 (0.8805, 0.9644) | 0.83 | 0.91 |
> | Ibot | Proposed Ultrasound Dataset | 90.85 | 0.9662 (0.9400, 0.9859) | 0.87 | 0.93 |
> | A-ViT | Proposed Ultrasound Dataset | 93.66 | 0.9742 (0.9481, 0.9930) | 0.95 | 0.94 |
>
> As suggested by the reviewer, we also evaluated Vision Transformer–based SSL baselines when pretrained on our ultrasound dataset instead of natural images. Table summarizes the results on breast cancer classification. Pretraining MAE, iBOT, and DINO on our ultrasound data improves their performance over ImageNet pretraining, and the proposed A-ViT with ACDT further achieves the highest accuracy and AUROC. The experiment confirms that (i) domain-aligned ultrasound pretraining is beneficial and (ii) the anatomy-aware design of A-ViT provides additional gains beyond non–anatomy-aware self-supervised methods.
>
> > Q3. Did the authors perform an ablation study to justify the anatomical-aware component? Showing results without the 'anatomy-aware' loss/mechanism would be critical evidence for the necessity and effectiveness of the proposed novelty.
> >
>
> **A4.**
>
> We agree that an explicit ablation of the anatomy-aware component is important to justify the novelty. Table 3 of the main manuscript provides an ablation on breast cancer classification that disentangles the effect of (i) ultrasound-specific pretraining, (ii) the anatomy conditioned deformable transformer (ACDT) module, and (iii) the adversarial and self-distillation objectives. Under the same ultrasound pretraining and MIM objective, adding the ACDT increases accuracy from 89.43% to 92.25% (+2.82%), demonstrating that adapting the deformable attention on anatomical context provides substantial gains beyond domain-aligned pretraining alone.
>
> > Q4. How are anatomical labels obtained for the 5.2M pretraining images? Are they derived from metadata, manual annotation, or automated prediction? If the latter, could label noise degrade representation quality?
> >
>
> **A5.**
>
> We appreciate the reviewer for the insightful question. For publicly available datasets, anatomical classes are explicitly defined at the dataset level, and for the newly curated data, anatomical labels are assigned using exam types and metadata, which minimizes the likelihood of misannotation.
>
> To further evaluate robustness, we trained A-ViT with synthetically injected anatomical label noise at rates of 5% and 20%. The results are summarized below:
>
> **Breast Cancer Classification [Fine-tuning] (A-ViT)**
>
> | Method | Noise Ratio | Accuracy | AUROC |
> | --- | --- | --- | --- |
> | A-ViT | 20% | 91.54 | 0.9624 |
> | A-ViT | 5% | 92.95 | 0.9709 |
> | A-ViT | 0% | 93.66 | 0.9742 |
>
> Although a moderate degradation in performance is observed as the level of injected noise increases, given that the true misannotation rate in our dataset is expected to be extremely low, we anticipate that the impact on real-world performance is expected to be minimal.

---

### Official Review · Reviewer_BKQr · 2025-10-31

**Soundness:** 3
**Presentation:** 2
**Contribution:** 3
**Rating:** 6
**Confidence:** 4

**Summary:**

This paper proposes an anatomy-aware representation learning framework (ARL) for medical ultrasound (US) imaging, based on a large-scale US dataset and a novel Anatomy-aware Vision Transformer (A-ViT). The method incorporates anatomical context via a deformable transformer and combines multiple self-supervised objectives to improve feature learning. Extensive experiments across multiple downstream tasks demonstrate improved performance over existing self-supervised learning baselines.

**Strengths:**

1.The paper introduces a large-scale, multi-source US dataset, which is a valuable contribution to the community.

2.The proposed A-ViT model effectively integrates anatomical information and shows consistent improvements across diverse US tasks.

3.The combination of MIM, adversarial loss, and self-distillation is well-motivated and empirically validated.

4.Comprehensive evaluation on multiple organs and tasks (classification and segmentation) strengthens the claim of generalizability.

**Weaknesses:**

1.The motivation for choosing certain design elements (e.g., deformable attention, specific loss weighting) could be better justified.

2.The comparison to other anatomy-aware or medical-specific transformers is limited.
[1]Anatomy-Aware Contrastive RepresentationLearning for Fetal Ultrasound
[2]Anatomy-Aware Self-Supervised Learning for Aligned Multi-Modal Medical Data
[3]SELF-SUPERVISED REPRESENTATION LEARNING FOR ULTRASOUND VIDEO

3.The computational cost and inference speed of A-ViT are not discussed, which may limit practical deployment.

4.Some baseline results (e.g., DINO v3) are strong, and the margin of improvement is not always substantial.

5.The main text lacks detailed statistical information about the dataset, such as organ, image size, depth, and classes.

**Questions:**

1.The authors are advised to check the title of the paper: “ANATOMY-AWARE REPRESENTATION LEARNING FOR MEDICAL ULTRASOUND.” There appears to be an extra symbol (a period at the end).

2.Why was the deformable attention mechanism chosen over other spatial-aware transformers? Were alternatives considered?

3.How does the model perform when the anatomical label is noisy or misassigned?

4.Is the performance gain mainly from the proposed architecture or the large-scale dataset? An ablation on dataset scale would be helpful.

---

> ### Author Response · Authors · 2025-11-20
>
> We sincerely thank Reviewer BKQr for the constructive and insightful feedback on our work. We have carefully addressed the comments below and incorporated the revisions into the manuscript.
>
> > [W1, Q2] The motivation and considerations for choosing certain design elements (e.g., deformable attention, specific loss weighting) could be better justified.
> >
>
> We appreciate the reviewer’s insightful comment regarding the need for clearer justification of the design choices, including deformable attention and loss weighting.
>
> **Anatomy conditioned deformable transformer (ACDT)**
>
> In medical ultrasound, the spatial distribution of visual patterns is inherently anatomy dependent. Imaging depth varies across target organs, and probe types and operating frequencies are selected according to the underlying anatomy. Since the characteristics of the organs and the measurement environment differ depending on the anatomical region being imaged, the spatial scale and location of diagnostically relevant features also vary accordingly. For instance, echocardiography relies on globally distributed structures such as cardiac chambers, whereas breast ultrasound primarily requires capturing localized features such as focal lesions.
>
> To explicitly account for these anatomy-specific characteristics, we propose that the model should adapt its effective receptive field according to the anatomy being imaged. Based on this motivation, we introduce the Anatomy-Conditioned Deformable Transformer (ACDT), which dynamically adjusts the receptive field and enhances the extraction of meaningful features under varying anatomical conditions.
>
> **Breast Cancer Classification**
>
> | Conditioning scheme | Accuracy | AUROC |
> | --- | --- | --- |
> | Cross attention | 90.84 | 0.9578 |
> | FiLM | 90.14 | 0.9624 |
> | LoRA | 92.25 | 0.9692 |
> | Proposed | 93.66 | 0.9742 |
>
> We also investigated alternative conditioning mechanisms, including LoRA [1], FiLM [2], and cross-attention–based conditioning [3], as potential ways to embed anatomy information into the transformer. The results for downstream breast cancer classification are summarized in the table. The proposed ACDT outperformed other conditioning schemes, demonstrating its effectiveness to capture anatomy-specific feature representations.
>
> **Adaptive loss weighting**
>
> To combine the masked-image modeling loss ($L_{\text{mim}}$) and adversarial loss ($L_{\text{ad}{\scriptsize\text{v}}}$), we employ an adaptive loss-weighting strategy rather than a fixed coefficient. MSE-based losses predominantly capture low-frequency structure, whereas adversarial losses better preserve high-frequency content [4]. Fixed weighting can lead to imbalanced optimization of the complementary image features. Adaptive weighting prevents domination by a single loss term and stabilizes representation learning.
>
> **Breast Cancer Classification**
>
> | Method | Loss weighting | Accuracy | AUROC |
> | --- | --- | --- | --- |
> | A-ViT | Constant weighting | 92.95 | 0.9701 |
> | A-ViT | Adaptive weighting | 93.66 | 0.9742 |
>
> Table 2 reports downstream A-ViT performance under constant and adaptive weighting. Adaptive weighting yields improved performance, supporting its effectiveness in balancing low- and high-frequency components during training.
>
> **References**
>
> [1] Hu, Edward J., et al. "Lora: Low-rank adaptation of large language models." ICLR 2022
>
> [2] Perez, Ethan, et al. "Film: Visual reasoning with a general conditioning layer." AAAI. 2018.
>
> [3] Vaswani, Ashish, et al. "Attention is all you need." Neurips, 2017.
>
> [4] Ledig, Christian, et al. "Photo-realistic single image super-resolution using a generative adversarial network." Proceedings of the IEEE CVPR 2017

---

> > ### Author Response · Authors · 2025-11-20
> >
> > > W2. The comparison to other anatomy-aware or medical-specific transformers is limited. (Anatomy-Aware Contrastive Representation, Learning for Fetal Ultrasound, Anatomy-Aware Self-Supervised Learning for Aligned Multi-Modal Medical Data, Self-Supervised Representation Learning for Ultrasound Video)
> > >
> >
> > **A2.**
> >
> > We agree that it is important to compare our approach with prior anatomy-aware or ultrasound-specific representation learning methods, and we appreciate the reviewer for pointing us to these relevant works. In the revised manuscript, we have expanded the related work section to clarify these distinctions.
> >
> > Jiao et al. propose a self-supervised framework that leverages fetal ultrasound videos to capture anatomical characteristics inherent to the fetal imaging domain [5]. Fu et al. further exploit fetal-specific anatomical cues by constructing anatomy-aware pretext objectives to enhance representation learning within the domain [6]. Hu et al. introduce a methodology that enforces anatomical consistency across multi-modal medical data, thereby preserving shared structural information across modalities [7].
> >
> > In contrast, our work aims to establish a representation learning framework applicable across a broad spectrum of ultrasound anatomies. We highlight the necessity of learning feature representations that explicitly account for the markedly different spatial distributions and structural patterns observed across anatomical regions. To this end, we curate a large-scale ultrasound dataset spanning 16 distinct anatomical categories and propose an Anatomy-Conditioned Deformable Transformer, designed to adapt its receptive fields and attention patterns based on the imaged anatomy.
> >
> > **References**
> >
> > [5] Jiao, Jianbo, et al. "Self-supervised representation learning for ultrasound video." IEEE, ISBI 2020.
> >
> > [6] Fu, Zeyu, et al. "Anatomy-aware contrastive representation learning for fetal ultrasound." ECCV, 2022.
> >
> > [7] Hu, Hongyu, et al. "Anatomy-Aware Self-Supervised Learning for Aligned Multi-Modal Medical Data." BMVC. 2022.
> >
> > > W3. The computational cost and inference speed of A-ViT are not discussed, which may limit practical deployment.
> > >
> >
> > **A3.**
> >
> > We appreciate the reviewer’s feedback regarding the practical feasibility of deploying A-ViT. To address this, we have supplemented results that reports the computational cost and inference speed of the standard Vision Transformer and our proposed A-ViT. Throughput was measured using 224×224 B-mode ultrasound images, and inference speed was evaluated on a single NVIDIA RTX 4090 GPU.
> >
> > | Model | Parameters | Flops | Inference speed |
> > | --- | --- | --- | --- |
> > | Vision Transformer | 86M | 17.58G | 12.1ms |
> > | A-Vit | 95M | 17.66G | 16.6ms |
> >
> > As shown in the table, A-ViT introduces only a marginal increase in parameters and computational overhead compared to the standard Vision Transformer, while providing substantially improved anatomy-aware representation learning.
> >
> > > W4. Some baseline results (e.g., DINO v3) are strong, and the margin of improvement is not always substantial
> > >
> >
> > **A4.**
> >
> > We acknowledge the reviewer’s observation that the margin of improvement for A-ViT may appear modest in certain individual downstream applications. We would like to emphasize that across all downstream tasks spanning diverse clinical indications, A-ViT consistently outperforms prior state-of-the-art (SoTA) methods, demonstrating strong generalizability and robustness.
> >
> > Importantly, as illustrated in Fig. 4 of the main manuscript, A-ViT exhibits a particularly pronounced advantage under challenging low-data scenarios. In the downstream task with only 0.4k training samples, A-ViT achieves a Top-1 accuracy of 74.5%, substantially outperforming the best competing method, DINO-v3 (Top-1 accuracy 65.81%).
> >
> > > W5. The main text lacks detailed statistical information about the dataset, such as organ, image size, depth, and classes.
> > >
> >
> > **A5.**
> >
> > In this paper, we constructed a large-scale ultrasound dataset consisting of approximately 5.2 million images sourced from 15 medical institutes across multiple countries, along with 11 publicly available datasets. The dataset includes images acquired using linear, convex, and sector array probes. The image pixel resolution ranges from 64×64 to 1280×960, with the height and width distributions characterized by 503.1 ± 167.1 and 655.8 ± 238.2 (mean ± std), respectively. The dataset further covers a broad span of imaging depths up to 24 cm. Anatomical categories comprise Abdominal, Aorta, Bladder, Breast, Cardiac, Doppler, Esophageal, Gallbladder, Kidney, Liver, Lung, Muscle, Oncology, Soft tissue, Testicle, Thyroid, and Others.
> >
> > We incorporated these dataset statistics into the revised manuscript to enhance clarity and completeness.

---

> ### Author Response · Authors · 2025-11-20
>
> > Q1. The authors are advised to check the title of the paper
> >
>
> **A6.** We sincerely thank the reviewer for pointing out the typographical error. The typo has been corrected in the revised manuscript.
>
> > Q3. How does the model perform when the anatomical label is noisy or misassigned?
> >
>
> **A7.**
>
> We appreciate the reviewer’s question regarding the model’s robustness to noisy or misassigned anatomical labels. In our dataset, the risk of label noise is inherently low. For publicly available datasets, anatomical classes are explicitly defined at the dataset level, and for newly curated data, anatomical labels are assigned using exam types and metadata, which minimizes the likelihood of misannotation.
>
> To further evaluate robustness, we trained A-ViT with synthetically injected anatomical label noise at rates of 5% and 20%. The results are summarized below:
>
> **Breast Cancer Classification**
>
> | Method | Noise Ratio | Accuracy | AUROC |
> | --- | --- | --- | --- |
> | A-ViT | 20% | 91.54 | 0.9624 |
> | A-ViT | 5% | 92.95 | 0.9709 |
> | A-ViT | 0% | 93.66 | 0.9742 |
>
> Although a moderate degradation in performance is observed as the level of injected noise increases, given that the true misannotation rate in our dataset is expected to be extremely low, we anticipate that the impact on real-world performance is expected to be minimal.
>
> > Q4. Is the performance gain mainly from the proposed architecture or the large-scale dataset? An ablation on dataset scale would be helpful.
> >
>
> **A8.**
>
> The large-scale ultrasound dataset proposed in this work, comprising a wide range of diagnostic categories, constitutes one of the key components enabling effective representation learning. The table below summarizes the performance degradation observed as the size of the self-supervised pretraining dataset is gradually reduced from 5.2M to 52K images. As demonstrated, sufficient volume and diversity of ultrasound data are essential for learning effective and generalizable representations, which contribute to improved downstream task performance.
>
> **Breast Cancer Classification**
>
> | Dataset Size | Accuracy | AUROC |
> | --- | --- | --- |
> | 5.2 M (100%) | 93.66 | 0.9742 |
> | 520 K (10%) | 90.84 | 0.9581 |
> | 52K (1%) | 87.23 | 0.9353 |

---

### Author Response · Authors · 2025-11-20
**General response**

We sincerely thank all the reviewers for their time and thoughtful feedback in reviewing our manuscript. In response to their valuable suggestions, we have carefully revised the manuscript and incorporated the following updates in the revised manuscript.

- **(Related work)** We have expanded the discussion comparing the proposed A-ViT with prior anatomy-aware and ultrasound-specific representation learning methods.
- **(Main text and Appendix A)** We have added detailed information about the proposed large-scale ultrasound dataset, as well as a description of the procedure used to annotate anatomy categories.
- **(Appendix C)** We have included an ablation study that varies the size of the ultrasound pretraining dataset to assess the importance of large-scale data for effective representation learning.
- **(Appendix D)** We have added an evaluation of existing self-supervised learning methods when pretrained on natural-image and ultrasound-image datasets.
- **(Appendix E)** We have compared the proposed ACDT with alternative conditioning mechanisms, including LoRA, FiLM, and cross-attention.
- **(Appendix F)** We have added an ablation study on the adaptive loss-weighting strategy.
- **(Appendix G)** We have incorporated comparisons of masked-image reconstruction using a standard MAE and the proposed A-ViT.
- **(Appendix H)** We have supplemented computational cost and inference speed analysis of the A-ViT.

We believe that these revisions have further enhanced the quality and completeness of the paper, and we hope that the updated version adequately addresses concerns raised by the reviewers.

---

### Meta-Review · Area_Chair_m49Y · 2026-01-06

**Summary:**

This submission was reviewed by three expert reviewers, with the ratings of: 2 borderline accept, and 1 borderline reject. The main concerns are around unclear motivation and unclear technical presentation, limited novelty and comparison to closely related works, practicality, marginal performance improvement, quantitative results, discrepancy between the claims and proposed techniques, some poor experimental performance, and other minor issues. The authors provided a rebuttal for the concerns raised by the reviewers, but no further response from the reviewers was presented, and there was no discussion.

After carefully going through all the review comments and the authors' rebuttal, it can be seen that most of the concerns and questions are addressed by the authors' further response and experiments. Although there are still some concerns remaining not fully addressed, the major concerns were cleared and the contributions made in this paper could benefit the sub-community and of interest to the sub-group of the ICLR audience. As a result, the AC is happy to recommend accept, but the authors need to incorporate their responses into the final version to make sure all the concerns are well addressed.

**Reviewer Concerns:**

Concerns that the AC thinks were addressed by the rebuttal: motivation for certain design elements; comparison to closely-related works is partially addressed; computational cost and inference speed; strong baseline concern; dataset stats; lack of clear description of the proposed ARL mechanism; lack of quantitative results; lack of evidence about the speckle-related distortions; the novelty concern is partially addressed; other minor issues.

Concerns that are still outstanding: more in-depth comparisons to those closely-related works is needed to be included in the revision; the significance of the proposed technical contributions and novelty are still a bit limited, when compared to those related works.

**Reviewer Scores:**

According to the review comments, and the rebuttal, for each review the reviewer might have changed their score in the way below, if they had been able to participate fully in the discussion:
* Reviewer BKQr: borderline accept to accept, or unchanged
* Reviewer h8d3:  borderline reject to borderline accept
* Reviewer cf7P: borderline accept to accept, or unchanged.

---

### Decision · Program_Chairs · 2026-01-26

Accept (Poster)